Tumor microenvironment related novel signature predict lung adenocarcinoma survival

Chen Juan
Zhou Rui zhourui2355@csu.edu.cn
Respiratory Medicine, The Second Xiangya Hospital of Central South University , Changsha, Hunan , China
Módos Dezső
Electronic publication date: 2021 Jan 14
Publication date: 2021
Volume: 9
Electronic Location ID: e10628
Received 2020 Jan 14; Accepted 2020 Dec 1
Copyright: © 2021 Chen and Zhou
Copyright year: 2021
Copyright holder: Chen and Zhou
License: This is an open access article distributed under the terms of the Creative Commons Attribution License, which permits unrestricted use, distribution, reproduction and adaptation in any medium and for any purpose provided that it is properly attributed. For attribution, the original author(s), title, publication source (PeerJ) and either DOI or URL of the article must be cited.
License URL: https://creativecommons.org/licenses/by/4.0/

Keywords: Lung adenocarcinoma, Tumor microenvironment, Prognosis, Overall survival, TCGA, GEO, Predictive gene signature

Funding: The authors received no funding for this work.

==============================
Background

Lung adenocarcinoma (LUAD) is the most common histological type of lung cancers, which is the primary cause of cancer‐related mortality worldwide. Growing evidence has suggested that tumor microenvironment (TME) plays a pivotal role in tumorigenesis and progression. Hence, we investigate the correlation of TME related genes with LUAD prognosis.

Method

The information of LUAD gene expression data was obtained from The Cancer Genome Atlas (TCGA). According to their immune/stromal scores calculated by the ESTIMATE algorithm, differentially expressed genes (DEGs) were identified. Then, we performed univariate Cox regression analysis on DEGs to obtain genes that are apparently bound up with LUAD survival (SurGenes). Functional annotation and protein-protein interaction (PPI) was also conducted on SurGenes. By validating the SurGenes with data sets of lung cancer from the Gene Expression Omnibus (GEO), 106 TME related SurGenes were generated. Further, intersection analysis was executed between the 106 TME related SurGenes and hub genes from PPI network, PTPRC and CD19 were obtained. Gene Set Enrichment Analysis and CIBERSORT analysis were performed on PTPRC and CD19. Based on the TCGA LUAD dataset, we conducted factor analysis and Step-wise multivariate Cox regression analysis for 106 TME related SurGenes to construct the prognostic model for LUAD survival prediction. The LUAD dataset in GEO (GSE68465) was used as the testing dataset to confirm the prognostic model. Multivariate Cox regression analysis was used between risk score from the prognostic model and clinical parameters.

Result

A total of 106 TME related genes were collected in our research totally, which were markedly correlated with the overall survival (OS) of LUAD patient. Bioinformatics analysis suggest them mainly concentrated on immune response, cell adhesion, and extracellular matrix. More importantly, among 106 TME related SurGenes, PTPRC and CD19 were highly interconnected nodes among PPI network and correlated with immune activity, exhibiting significant prognostic potential. The prognostic model was a weighted linear combination of the 106 genes, by which the low-OS LUAD samples could be separated from the high-OS samples with success. This model was also able to rebustly predict the situation of survival (training set: p-value < 0.0001, area under the curve (AUC) = 0.649; testing set: p-value = 0.0009, AUC = 0.617). By combining with clinical parameters, the prognostic model was optimized. The AUC achieved 0.716 for 3 year and 0.699 for 5 year.

Conclusion

A series of TME-related prognostic genes were acquired in this research, which could reflect immune disorders within TME, and PTPRC and CD19 show the potential to be an indicator for LUAD prognosis and tumor microenvironment modulation. The prognostic model constructed base on those prognostic genes presented a high predictive ability, and may have clinical implications in the overall survival prediction of LUAD.

Introduction

Lung cancer is the most common cause of cancer-related deaths worldwide (Chen et al., 2016). Non-small cell lung cancer (NSCLC) represents 85% of lung cancers, mainly including lung adenocarcinoma (LUAD), lung squamous cell carcinoma (LSCC) (Losanno & Gridelli, 2016). Notably, the incidence of LUAD significantly increased and surpassed of LSCC and it constitutes nearly 40% of all lung malignancies (Nakamura & Saji, 2014). Although with multiple kinds treatment methods, including surgery, chemotherapy, radiotherapy, and dramatic treatment shift of some targeted therapeutic agents, the prognosis for LUAD patients remains poor worldwide, with 5-year relative survival currently at 18% (Siegel, Miller & Jemal, 2015; William et al., 2009; Shaw et al., 2011). Therefore, understanding the mechanism of carcinogenesis and therapeutics of lung cancer is quite important.

At present, the pathogenesis of LUAD has not been adequately described; meanwhile, there are increasing study support the view of tumor microenvironment critically influence gene expression of tumor tissues, and the clinical outcomes further. The tumor microenvironment (TME) is a complicated mixture, immune and stromal cells are two major types of non-tumor factors. They have been certified to promote the development of diagnostic and prognostic assessment process of lung cancer (Cui et al., 2015; Zhan et al., 2015; Zhao, Li & Tian, 2018). Hence, exploring the molecular composition and function of the TME is critical to effectively manage cancer progression and immune response (Wood et al., 2014; Li et al., 2017).

Previous studies have solved the problem of the complexity of tumor infiltrating immune cells in LUAD. Most of these studies assessed tumor-infiltrating immune cells by immunohistochemical analysis of a single marker (Mansuet-Lupo et al., 2016; Driver et al., 2017; Connolly, Nader & Joshi, 2018). Fortunately, computational analysis of tumor immune cell interactions is now available by bioinformatics tools. In 2013, Yoshihara firstly reported an algorithm called ESTIMATE (Estimation of STromal and Immune cells in MAlignant Tumor tissues using Expression data). The infiltration level of stromal and immune cells was predicted by calculating stromal and immune score based on data from the Cancer Genome Atlas (TCGA) data sets by ESTIMATE (Yoshihara et al., 2013). Subsequent reports applied the innovative algorithm to different cancer, such as prostate cancer (Shah et al., 2017), breast cancer (Priedigkeit et al., 2017), and colon cancer (Alonso et al., 2017), which further confirmed the effectiveness of the algorithm based on big database.

We analyzed immune/stromal score of LUAD cohorts from TCGA, which was derived from ESTIMATE algorithm, and extracted a series of TME associated prognostic gene in LUAD. Among those TME associated prognostic gene, PTPRC and CD19 were of particular interest for they are highly interconnected in the PPI network, and closely correlated with immune-related activity by GSEA analysis and CIBERSORT analysis. Combining those prognostic genes, we constructed a prognostic model which could provide a moderate OS prediction for LUAD alone, and provide a robust prediction with clinical parameters.

Materials and Methods

Data acquisition

A total of 576 samples, gene expression RNAseq and clinical information such as pathological stage, survival of LUAD were downloaded from the TCGA database and prepared for the analysis of differential expression. The data set was submitted by University of North Carolina TCGA genome characterization center based on the Illumina HiSeq 2000 RNA Sequencing platform (Oct 13, 2017). We downloaded the estimate, immune and stromal score of 517 LUAD samples from TCGA in ESTIMATE website. Stromal score captures the presence of stroma in tumor tissue, immune score represents the infiltration of immune cells in tumor tissue, and estimate score that infers tumor purity. The estimate score is equal to the immune score plus the stromal score. Fifty-nine samples were deleted for lack of immune and stromal score in ESTIMATE website. Gene expression profiles and LUAD clinical covariates (stage, age, sex) were obtained from GEO database with the accession number GSE68465, consisting of 442 samples (Shedden et al., 2008). After using the Robust Multichip Averaging (RMA) method to normalize the sequence matrix data, the data was used to validate the differential expression gene and test the prognostic model. The data set was submitted by Mervi Heiskanen based on Affymetrix HG-U133A (May 01, 2015) (Marti et al., 2003). A total of 133 non-small cell lung cancer samples were downloaded from GEO database with the accession number GSE14814, were to compare the prognostic model with other gene expression signature.

Distinguishing of differentially expressed genes

The raw data of TCGA was pre-processed by limma algorithm (Ritchie et al., 2015). The adjusted p-values (adj. p) < 0.05 and |Log2 (FC)| > 1 were set as the cut-offs to screen for differentially expressed genes (DEGs).

Heatmaps and clustering analysis

ClustVis web tool was used to create heatmaps and clustering (Metsalu & Vilo, 2015).

Enrichment analysis of DEGs

DAVID conducted functional enrichment analysis, including biological process, molecular function and cell component to analyze DEGs (Huang, Sherman & Lempicki, 2009), an essential foundation for visualization, annotation, and integrated discovery. KEGG (Kyoto Encyclopedia of Genes and Genomes) pathways was also performed by the database of DAVID. Whole genome was the background using DAVID. False discovery rate (FDR) < 0.05 was used as the cut-off.

Screening of survival-related DEGs

Kaplan–Meier plots and univariate Cox regression were used to visualize the association between the genes expression and overall survival of patients to explore its prognostic value. Statistical significance was examined using the Log-rank test. p < 0.05 were considered statistically significant.

Integration of PPI network and screening of modules

Search Tool for the Retrieval of Interacting Genes (STRING) online database was used to retrieve predicted PPIs. Only interactions with a combined score >0.4 of all associations obtained in STRING were selected to construct the PPI network using Cytoscape software (Bader & Hogue, 2003). The Molecular Complex Detection (MCODE) plugin in Cytoscape was then utilized to find clusters of the PPI network. Top 8 nodes ranked by degree were identified as hub genes.

Gene set enrichment analysis

From Molecular Signatures Database, Hallmark and C7 gene sets v6.2 collections were downloaded as the target sets. GSEA performed using the software gsea-3.0. 517 samples from TCGA were used for GSEA, and only gene sets with nominal (NOM) p-value < 0.05 and FDR < 0.25 were considered as significant.

TICs profile

CIBERSORT computational method was applied for estimating the tumor-infiltrating immune subsets (TICs) abundance profile in all tumor samples, which followed by quality fifiltering that only 517 tumor samples with p < 0.05 were selected for the following analysis.

Construction of the risk assessment model

To reduce the dimensionality and eliminate collinearity, factor analysis was performed on 106 prognostic genes. Then we used the Step-wise multivariate Cox regression analysis to obtain factors correlated with overall survival significantly (SurFactors). By combining the score coefficient of factors weighted by their regression coefficients and standard deviation, the risk index of each gene was calculated as follows: αi=∑j=1mβj×facijstdi

where αi was the risk index of gene i, m the number of SurFactors, βj the regression coefficient of factor j in multivariate Cox regression analysis, facij the factor score coefficient of j-th factor over gene i and stdi the standard deviation of gene i.

By combining the expression values of prognostic genes weighted by their risk index, the following risk scores can be established for each patient: Risk_score=∑i=1nexpi∗αi

where n was the number of prognostic genes, and expi the expression value of gene i.

Survival and statistical analysis

We calculated risk score (RS) for every sample from testing data set based on the prognostic model and divided the specimens into two different groups based on the median RS. Kaplan–Meier survival curves were drawn and compared the two subgroups via log-rank tests. We divided the samples into two group according to their overall survival, receiver operating characteristic (ROC) curves were drawn by IBM SPSS statistics 22, and area under the curve (AUC) was calculated. Moreover, the univariate and multivariate analyses of survival were conducted to identify the prognostic factors for LUAD patients from TCGA data set. A nomogram and Calibration plots were established based on the TCGA LUAD cohort. The nomogram and calibration plot analysis were conducted by using the R package “rms” and “rmda.” All tests were two-tailed, and p < 0.5 was considered statistically significant. All statistical analyses were conducted using R software version 3.4.2.

Results

ESTIMATE algorithm of LUAD

Transcriptional expression profiles and clinical information of 517 LUAD patients were collected from TCGA database. Among them, 277 (53.6%) patients were female, 240 (46.4%) were male. Pathological stage included 277 patients (53.6%) of stage I, 122 (23.6%) stage II, 84 (16.3%) stage III, 26 (5%) stage IV and 8 (1.5%) unknown. Based on ESTIMATE algorithm, the scores of stromal and immune were calculated, ranging from −1355.85 to 3286.67 and −1959.31 to 2098.77, respectively. The Estimate score was significantly associated with Pathological stage (Fig. 1A, p = 0.0436). The lowest Estimate score was at the most advanced pathological stage, the stage IV. Immune score showed remarkable prognostic potential, correlated with the pathological stage (Fig. 1B, p = 0.026), while stromal score showed no correlation with the pathological stage (Fig. 1C, p = 0.145).

Figure 1 The relationship between Estimate/immune/stromal score and the prognosis of LUAD samples from TCGA.

(A) Estimate score was significantly associated with pathological stage (p = 0.0436). (B) Immune score showed predictive potential to pathological stage (p = 0.026). (C) Similarly, the lowest stromal score was found in the most progressive clinicopathological stage IV, however, it was not statistically significant (p = 0.145). (D) Immune score was significantly correlated with overall survival of LUAD samples (p = 0.015). (E) Stromal score was not statistically significant correlated with overall, but the median overall survival of cases with higher stromal scores also showed longer than the patients with lower stromal scores (1830 d vs 1293 d, p = 0.0599, p = 0.145).

To explore whether the potential correlation existing between survival benefits and immune/stromal score, 517 LUAD patients were divided into high and low score groups based on their scores. Kaplan–Meier survival curves (Fig. 1D) showed that immune score significantly correlated with overall survival. Compared to the cases in the low score group, patients with high immune score have longer median overall survival (1725 d vs 1235 d, p = 0.0152 in log-rank test). Although there was no statistically significant correlation between the stromal score and overall survival, patients with higher stromal score had longer median overall survival (Fig. 1E, 1830 d vs 1293 d, p = 0.0599 in log-rank test).

Identification of DEGs

To identify DEGs profiles according to immune/stromal score, we obtained the gene expression array of 517 LUAD patients obtained TCGA database. Based on the comparison of immune score of high group and low group, after analysis with the limma software package algorithm, there were 903 genes up-regulated and 56 genes down-regulated. We draw the heatmap with the top highly variate gene to (Fig. 2A). Similarly, 1,007 up-regulated genes and 30 down regulated genes were obtained for high stromal score compared with low score (Fig. 2B). It is obvious that up-regulated DEGs, no matter for comparison based on immune score nor stromal score, take the major part in total DEGs. Therefore, we decide to focus on up-regulated DEGs in further analysis.

Figure 2 Differential expression genes derived from the comparision of immune/stromal high score group vs low score group.

(A and B) Heatmap of top 50 highly variaed gene from immune score/stromal score group. Genes with higher expression are shown in red, lower expression are shown in green, genes with same expression level are in black. (C and D) Venn diagram of common up/down regulated of immuneDEGs and stromal DEGs.

Enrichment analysis of DEGs

To reveal the biological function of the DEGs, GO and KEGG pathway enrichment analysis were performed for the up-regulated DEGs. GO analysis showed that immune up-regulated DEGs were remarkably enriched in immune response by biological processes, receptor activity by molecular functions, and plasma membrane by cellular component, respectively (Figs. 3A–3C). Similarly, stromal up-regulated DEGs mainly enriched in cell adhesion by biological processes, extracellular matrix binding by molecular functions, and extracellular space by cellular component, respectively (Figs. S1A–S1C). KEGG pathway were both mainly enriched in cytokine-cytokine receptor interaction and cell adhesion molecules (CAMs) pathway (Fig. 3D; Fig. S1D).

Figure 3 GO term and KEGG pathway analysis for immune up-regulated DEGs.

Top 10 pathways. False discovery rate (FDR) of GO analysis was acquired from DAVID functional annotation tool. p < 0.05. (A) biological process, (B) cellular component, (C) molecular function, (D) KEGG pathway.

Overall survival analysis of DEGs

To make further efforts to elucidate if up-regulated DEGs could give benefits to the LUAD patient survival, Kaplan–Meier survival and univariate Cox proportional hazards regression analyses were performed on the up-regulated DEGs. The cases whose overall survival <30 days were excluded. The results showed that 446 immune related DEGs correlated with patient survival (p < 0.05), 387 stromal related DEGs correlated with patient survival (p < 0.05). Among all those genes, there were 291 duplicates of immune and stromal genes, a total of 542 genes (prognostic DEGs) associated with overall survival in patients with LUAD (Table S1).

Construction of PPI network by prognostic gene

Based on the STRING database, PPI network was obtained by using Cytoscape software to clarify the interaction between prognostic DEGs. The network was constructed by 13 modules, and we choose the top three significant modules ranked by mcode score for further study (Fig. 4). A total of 527 nodes and 4,661 edges were screened from the PPI network. According to the criteria above, eight nodes (PTPRC, ITGAM, LCP2, CTLA4, CD80, ITGAX, CD19, CCR5) were identified as hub genes. Consistently, these hub genes serve crucial roles in maintaining the top three modules.

Figure 4 PPI network of top three modules, ranked by mcode score.

(A–C) The top three significant modules ranked by mcode score. The color of a node in the PPI network reflects the number of interacting proteins with the designated protein, and the size of node indicates the log (FC) value of the Z score of gene expression.

Prognostic gene validate and intersection analysis with PPI network

To test whether these prognostic genes have prognostic significance in other LUAD cases. Microarray expression profile dataset GSE68465 from the GEO database was downloaded and the data were subjected to prognostic gene selection. There are 176 genes among 542 prognostic genes cannot find in GSE68465 dataset because of platform differences. Of the remaining 366 genes, 106 genes were confirmed involving in LUAD patient survival. By intersection analysis the 106 prognostic genes with hub genes of PPI network, only PTPRC and CD19 were associated with overall survival in patients with LUAD.

The correlation of PTPRC and CD19 expression with clinicopathological factors

Based on the study above, PTPRC and CD19 expression level were correlated with overall survival of LUAD patient, and PTPRC and CD19 high-expression group with longer survival. To explore the correlation of PTPRC and CD19 expression with clinical characteristic in LUAD patient, we analysis the PTPRC and CD19 expression level with TNM stage. The result indicated that the PTPRC and CD19 expression were negative correlated with the TNM stage of LUAD patient, and with the TNM stage rising, the expression of PTPRC and CD19 declined (Fig. 5).

Figure 5 The correlation of PTPRC and CD19 with clinicopathological stage and survival of LUAD patients.

(A and B) The correlation of PTPRC and CD19 expression with clinicopathological stage. ANOVA served as the statistical significance test. (C and D) Survival analysis for LUAD patients with different PTPRC and CD19 expression. Patients were labeled with high expression or low expression depending on the comparison with the median expression level. Log-rank test served as the statistical significance test.

PTPRC and CD19 could be an indicator in TME status

To further elaborate the role of PTPRC and CD19 in LUAD, 517 samples from TCGA were divined into high/low expression group based on median expression. We implement GSEA in the high and low expression group of PTPRC and CD19. The result display that, in the high expression level whether PTPRC or CD19, the genes were basically enriched in immune relative activities including allograft rejection, complement and inflammatory response. In the low expression level of PTPRC, the genes were basically enriched in glycolysis, and typical tumor pathway including MYC-targets-V1 and MYC-targets-V2. As to the low expression group of CD19, genes were gathered in metabolic pathway. In C7 collection, many genes were concentrated in the high expression group of no matter PTPRC nor CD19, and both low expression group enrich few gene (Fig. 6).

Figure 6 GSEA for samples with high PTPRC/CD19 expression and low expression.

(A) The enriched gene sets in HALLMARK collection by the high PTPRC expression sample. Up-regulated genes on the y-axis approaching the origin of the coordinates; by contrast, the down-regulated lay on the x-axis. Only gene sets with NOM p < 0.05 and FDR q < 0.25 were considered signifificant. Several leading gene sets were displayed in the plot. (B) The enriched gene sets in HALLMARK by samples with low PTPRC expression. (C) Enriched gene sets in C7 collection, the immunologic gene sets, by samples of high PTPRC expression. Several leading gene sets are shown in plot. (D) Enriched gene sets in C7 by the low PTPRC expression. (E) The enriched gene sets in HALLMARK collection by the high CD19 expression sample. (F) The enriched gene sets in HALLMARK by samples with low CD19 expression. (G) Enriched gene sets in C7 by the high CD19 expression.

Further, we performed correlation analysis between PTPRC and CD19 and 22 kind of immune infiltration cell. The result reveal that 16 kinds of immune infiltration cell were related with the expression of PTPRC (Fig. 7), 15 kinds associated with CD19 (Fig. S2).

Figure 7 TIC profifile in tumor samples and correlation of TICs proportion with PTPRC expression.

(A) Barplot showing the proportion of 22 kinds of TICs in LUAD tumor samples. Column names of plot were sample IDs. (B) Heatmap showing the correlation between 10 kinds of TICs and numeric in each tiny box indicating the p value of correlation between two kinds of cells. The shade of each tiny color box represented corresponding correlation value between two cells, and Pearson coeffificient was used for significance test. (C–P) Scatter plot showed the correlation of 16 kinds of TICs proportion with the PTPRC expression (p < 0.05). The red line in each plot was fitted linear model indicating the proportion tropism of the immune cell along with PTPRC expression, and Pearson coeffificient was used for the correlation test.

Construction and validation of the prognostic model

Factor analysis was used to determine common axes (or dimensions) of patterns and structures, which were measured by a reduced set of 106 predicted genes. The first twelve factors explained about 81.808% of the variation in the 106 prognostic genes. After multivariate Cox regression analysis, five factors correlated with survival were obtained (p < 0.05). Through the formula mentioned before, the final prognostic model combined with 106 candidate gene were constructed. Further, to test the predictive ability of the prognostic model, the microarray expression profile dataset GSE68465 was downloaded and we calculate the risk score for each patient. Based on the threshold risk score, we divided the patients into high- and low-risk group. We choose the log-rank test to identify the differences of OS between subgroups. The results showed that the overall survival of the high-risk patients is shorter than those in the low-risk ones in both TCGA dataset and GSE68465 (p < 0.0001, p = 0.0002 respectively, Figs. 8A and 8B). Then, to estimate whether the prognostic model is predictive of relapse free survival (RFS). We divided the patients into high- and low-risk group compared with the median risk score, log-rank test indicate that the prognostic model we build could predict the RFS of LUAD. ROC curves were also applied to evaluate the sensitivity of the prognostic model, and the ROC curves showed that the AUC value of the prognostic model reached 0.649, 0.617 in TCGA dataset and GSE68465 respectively. The result indicates a substantially effective performance of the prognostic model for overall survival prediction (Figs. 8C and 8D).

Figure 8 Validation of the prognostic model.

(A) The Kaplan–Meier survival analysis of the prognostic model for LUAD samples from TCGA. (B) ROC curve of the prognostic model in LUAD samples from TCGA. (C) The Kaplan–Meier survival analysis of the prognostic model for LUAD samples from GSE68465. (D) ROC curve of the prognostic model in LUAD samples from GSE68465. (E) The Kaplan–Meier survival analysis of the prognostic model with relapse free survival for LUAD samples from TCGA.

Comparison with other gene expression signatures

We compared the prognostic model constructed by 106 candidate gene with 9-gene biomarker, as it also use estimate score on TCGA datasets to build the prognostic model (Chen et al., 2020). GSE14814 included all 106 genes of our prognostic model and 9 genes needed to constructed the 9-gene biomarker. For GSE14814 data set, the 106-gene prognostic model achived a higher C-index (C-index, 0.56; 95% CI [0.48–0.64]; p < 0.01) compared with the 9-gene biomarker (C-index, 0.50; 95% CI [0.44–0.56]; p = 0.7) (Fig. S3).

To optimize the model with clinical charateristics

Further, univariate and multivariate Cox regression analysis have been executed between some of clinical pathological parameters and risk score of the prognostic model (Table 1). The result showed that the risk assessment model was an independent prognostic factor for prognostic. As the constructed risk assessment model with great prognostic value, we intended to improve the prognostic accuracy by intergrating with LUAD clinicopathological factors. We designed a nomogram to predict the survival of LUAD patient by combing T stage, lymph nodes metastasis, recurrence and risk score (Fig. 9A). The AUC of the model was achieved 0.716 for 3 year and 0.699 for 5 year (Figs. 9D and 9E). Figures 9C and 9D show the nomogram calibration plots for predicting the overall survival of 3 years and 5 years of patient.

Table 1 The univariate and multivariate Cox regression analysis of the patients from The Cancer Genome Atlas (TCGA).

Parameters	Univariate	Multivariate	
	HR	95% CI	p-value	HR	95% CI	p-value	
Recurrence (yes vs no)	2.405	[1.709–3.385]	<0.001	2.449	[1.738–3.450]	<0.001	
Age (≤65y vs >65y)	1.157	[0.860–1.558]	0.336				
Gender (male vs female)	1.047	[0.782–1.400]	0.759				
Risk score	2.216	[1.635–3.004]	<0.001	1.897	[1.329–2.709]	<0.001	
T stage	2.455	[1.697–3.553]	<0.001	1.827	[1.827–1.156]	<0.01	
N stage	2.546	[1.900–3.411]	<0.001	2.139	[1.517–3.017]	<0.001	
M stage	1.027	[0.747–1.414]	0.868				
TNM stage (I-II vs III-IV)	2.686	[1.973–3.658]	<0.001	1.963	[1.579–3.141]	0.155	
Number-pack-years-smoked	1.028	[0.716–1.475]	0.881				

Figure 9 Nomogram and Calibration curve.

(A) Construction of a nomogram for predicting survival probability at 3, 5, and year of LUAD cases from TCGA data set. Calibration curve for the nomogram when predicting 3 (B) and 5 (C) year OS. (D–E) ROC curve of the optimized model in 3-years and 5 years.

Discussion

Lung-cancer-related deaths is quite a large scale in the world which could be the most in patients who suffer cancer. And LUAD, as the most common type of lung cancer, is able to account for more than half of the morbidity and mortality of the lung cancer patient (Travis et al., 2011; Jemal et al., 2011). Currently, growing evidence has suggested that TME plays a pivotal role in tumor initiation and progression (Chen et al., 2020; Liotta & Kohn, 2001), especially the recent immunocheckpoint inhibitors have noticeable effects on lung adenocarcinoma. However, patient prognosis and disease progression involved with TME related genes in LUAD have not been elucidated clearly. In the present study, 1,910 genes involved in immune response and cell adhesion were identified by comparing different immune/stromal score of LUAD samples from the TCGA database. Besides, we performed survival analysis and revealed that 542 of them were associated with overall survival in LUAD patients. After performing cross validation through GSE68465, we obtained 106 TME related genes in which prominent correlation is found compared with prognosis situation. Of these survival-associated gene, 59 genes have been reported with the preliminary result. More importantly, among 106 TME related SurGenes, PTPRC and CD19 were highly interconnected nodes among PPI network and correlated with immune activity, exhibiting significant prognostic potential. Meanwhile, to explore the significance of these genes simultaneous changes for LUAD, we constructed and validated a risk assessment model that predicted survival of LUAD based on 106 genes.The prognostic model achieved robust predictive ability by combing with the clinical parameter we filtrated by multivariate Cox regression analysis.Recently, we notice that there are other prognostic model build by immune gene in LUAD (Chen et al., 2020), however, compared with their prognostic model, the prognostic model we build have some advantages. First of all, we constructed the model by factor analysis. Factor analysis is a kind of algorithms in biometrics. It represents a complex array of structure-analyzing procedures used to identify the interrelationships among a large set of observed variables and then, through data reduction, to group a smaller set of these variables into dimensions or factors that have common characteristics. It is a tool to reduce multidimensional data to lower dimensions while retaining most of the information (Barber, Faure & Long, 2004). Secondly, our prognostic model was constructed by immune gene and stromal gene rather than only included immune gene, considering the effect of stromal gene on tumor microenvironment. Thirdly, before used to constructed the model, these 106 genes have been performed univariate cox regression analysis, and all of these gene were correlated with the prognosis of LUAD. At last, our prognostic model is also predictive not only to overall survival, but also to relapse free survival.

Among 106 TME related genes to be associated with LUAD, we are especially interested in PTPRC and CD19, as in the PPI network, they are highly interconnected nodes. PTPRC is a member of the protein tyrosine phosphatase (PTP) family. PTPs are known to be signaling molecules that regulate a variety of cellular processes including cell growth, differentiation, mitosis, and oncogenic transformation. PTPRC has been shown to be an essential regulator of T- and B-cell antigen receptor signaling. It functions through either direct interaction with components of the antigen receptor complexes, or by activating various Src family kinases required for the antigen receptor signaling. PTPRC also suppresses JAK kinases, and thus functions as a regulator of cytokine receptor signaling. Alternatively spliced transcripts variants of this gene, which encode distinct isoforms, have been reported. There were studies showed the presence of intraepithelial PTPRC + cells from NSCLC or SCLC patients predicts favorable disease-specific survival (Kilvaer et al., 2016; Wang et al., 2013). However, the mechanism of PTPRC in lung cancer has not been elucidated. The GSEA enrichment analysis in our research showed the genes in PTPRC high-expression group were mainly enriched in immune-related activities, such as allograft rejection, complement, and interferon response, and enriched in metabolic pathways in low expression of them, including glycolysis, oxidative phosphorylation, and typical tumor pathways. These result indicate that the expression level of PTPRC might correlate with the status of immune and microenviroment in tumor. Further study analysised by CIBERSORT algorithm showed many tumor infiltrating immune cell were correlated with the expression of PTPRC. All of these result suggest PTPRC participating in tumor microenvironment, and could be an indicator of TNM status. CD19 functions as coreceptor for the B-cell antigen receptor complex (BCR) on B-lymphocytes. Decreases the threshold for activation of downstream signaling pathways and for triggering B-cell responses to antigens (De Rie et al., 1989; Carter & Fearon, 1992; Van Zelm et al., 2006). Required for normal B cell differentiation and proliferation in response to antigen challenges (Carter & Fearon, 1992; De Rie et al., 1989). Some studies had reported that CD19 might serve as a downstream effector in RET signaling pathway and EGFR-activated signals in NSCLC. Other research have found in subcutaneous tumor of mice injected with LP07 adenocarcinoma cells and then treated with CD19 monoclonal antibodies, compared to non-treated cancer mice, in tumors of monoclonal-treated animals, tumor area and weight and ki-67 were significantly reduced, which implied that CD19 seemed to play a protumoral role in LUAD. There were researches reported the bronchoalveolar lavage fluid subset composition in primary lung neoplasia differs from that in the peripheral blood through an increase in CD19+ B cells compared with the peripheral blood. Similarly, some studies have described CD19+ B cells as the pathogenic populations within lung tumors (Tumor-infiltrating B cells: their role and application in anti-tumor immunity in lung cancer, Bal T lymphocyte subsets are reduced in primary lung neoplasias) (Wang et al., 2019; Cascio et al., 1993). However, Our results suggested that the expression of CD19 was decreased in the advancing stages of LUAD patients. According to another study, in non small cell lung cancer, CD19+ B cells decrease by cancer disease (Biomarkers related to immunosenescence: relationships with therapy and survival in lung cancer patients) (Saavedra et al., 2016). Thus, CD19 might play a double-face role in tumor, in different stages of tumor, either promoting survival or inducing apoptosis. The GSEA results showed that immune-related signaling pathways, such as allograft rejection, complement, and interferon response, were significantly enriched in the CD19 high expression group. In the CD19 low-expression group, metabolic pathways including adipogenesis, cholesterol homeostasis, and typical tumor pathways were enriched. These results implied that CD19 might participate in the status conversion of TME from immune-dominant to metabolic-dominant. Further analysis of TIC supported this view. Accordingly, the downregulation of CD19 along with the advancing stage of LUAD, the conversion of TME from immune-predominant to metabolic-dominant status, and the reduction of antitumor TICs supported that CD19 might play an antitumor role in LUAD.

Conclusions

In summary, after analysis of immune/stromal score by using the ESTIMATE algorithm in TCGA database, a series of TME related genes was obtained and validated by an independent LUAD cohort. Further, the prognostic model combined with those identified genes was constructed and proved could provide a moderate OS prediction for LUAD. By integrating with clinical characteristics, testing by multivariate Cox regression analysis, the predictive ability have been optimized. Among 106 TME related genes, PTPRC and CD19 are highly interconnected nodes in the PPI network. Meanwhile, bioinformatic evidence suggest the levels of them affected the immune activity of TME. These two prognostic genes might be a candidate target to monitor and treat lung cancer, attributing to their immune nature and prognostic significance, thereby improving the clinical outcomes. Finally, our study might provide a novel insight on the potential correlation between monitoring and manipulating the TME with LUAD prognosis and precision immunotherapies.

Supplemental Information

Supplemental Information 1 GO term and KEGG pathway analysis for stromal up-regulated DEGs.

Top 10 pathways. False discovery rate (FDR) of GO analysis was acquired from DAVID functional annotation tool. p <0.05. (A) biological process, (B) cellular component, (C) molecular function, (D) KEGG pathway.

Click here for additional data file.

Supplemental Information 2 C-index Comparison Between 14-gene biomarker and 106-gene biomarker.

Comparison of C-index between 14-gene biomarker and 106-gene biomarker in GSE14814 data set.

Click here for additional data file.

Supplemental Information 3 Scatter plot showed the correlation of 15 kinds of TICs proportion with the CD19 expression (p < 0.05).

The red line in each plot was fifitted linear model indicating the proportion tropism of the immune cell along with CD19 expression, and Pearson coeffificient was used for the correlation test.

Click here for additional data file.

Supplemental Information 4 Cox coefficients and significance of cox regression.

Click here for additional data file.

Additional Information and Declarations

Competing Interests

Author Contributions

Data Availability

The authors declare that they have no competing interests.

Juan Chen performed the experiments, analyzed the data, prepared figures and/or tables, and approved the final draft.

Rui Zhou conceived and designed the experiments, authored or reviewed drafts of the paper, and approved the final draft.

The following information was supplied regarding data availability:

Data used for the differential expression gene analysis and prognostic model construction is available at TCGA: https://xenabrowser.net/datapages/?dataset=TCGA.LUAD.sampleMap%2FLUAD_clinicalMatrix&host=https%3A%2F%2Ftcga.xenahubs.net&removeHub=https%3A%2F%2Fxena.treehouse.gi.ucsc.edu%3A443 and https://xenabrowser.net/datapages/?dataset=TCGA.LUAD.sampleMap%2FHiSeqV2&host=https%3A%2F%2Ftcga.xenahubs.net&removeHub=https%3A%2F%2Fxena.treehouse.gi.ucsc.edu%3A443.

Data used to validate the differential expression gene and the prognostic model is available at NCBI GEO: GSE68465.

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
