# Peer review of "Tumor microenvironment related novel signature predict lung adenocarcinoma survival"

_PeerJ, doi:10.7717/peerj.10628_

## Round 0.1 · original submission · Major Revisions

After the author's appeal, I have reconsidered my decision.

I have checked both papers which are mentioned the submitted one and the recently published Chen et al (doi.org/10.1002/jcb.29667). The Chen et al paper was published after the manuscript was submitted.

The authors of the manuscript have used at the first step the ESTIMATE method to make subgroups of the TCGA samples, similarly as the Chen et al. The Chen et al paper uses only the immune score meanwhile the submitted paper uses both immune and stromal score to distinguish and find differentially expressed genes and build up a signature. However, only the immune score distinguished patient survival in the data. The added novel results are the slight difference in the analysis of the gene signatures, the authors risk score and the network analysis. The latter one has a few flaws which the reviewers mentioned. The risk score's AUC (0.649) shows some predictive power, but it is barely relevant in any clinical settings.

After considering all of these I think the paper can have a major revision but needs a thoughtful analysis by the authors comparing the Chen et al paper what this manuscript The revised manuscript also needs to address the statistical problems and co-founding factors which the reviewers mentioned as well.

· Appeal

Appeal


· · Academic Editor

Reject

After careful consideration, I have to agree with reviewer 3. The manuscript does not hold sufficient additional information compared to Chen et al, doi.org/10.1002/jcb.29667. The manuscript uses very similar datasets and methods for the reanalysis. Besides, according to the reviewers, the statistical analysis of the manuscript is insufficient as well. The provided network analysis does not add extra scientific information, compared to Chen et al.

Due to the above-mentioned reasons, I regret to say but I reject the manuscript.

Sincerely yours,

Dr Dezso Modos

Reviewer 1 ·

Basic reporting

1.1 Omission of key prior publications is a weakness of the current manuscript. Additional papers should have been discussed in the existing relevant reportings.
Eg.
[1] Chen G, Dong Z, Wu D, Chen Y. Profiles of immune infiltration in lung adenocarcinoma and their clinical significant: A gene-expression-based retrospective study [published online ahead of print, 2020 Jan 31]. J Cell Biochem. 2020;10.1002/jcb.29667. doi:10.1002/jcb.29667.

1.2 In figure 1 to 8, organization of all charts needs to be rearranged. The charts should be align and sufficient resolution. The content of main text should be corresponding with the results in charts.

Experimental design

2.1 “Construct and validation the prognostic model” Line 236, “…the final prognostic model combined with 106 candidate gene were constructed…” 106 TME related SurGenes is might the most important innovation point of this research. Thus, Multivariate Cox regression analysis should be further preformed between age, gender, TNM stage and Risk score in this model. The author may refer to this article:
[1] Chen G, Dong Z, Wu D, Chen Y. Profiles of immune infiltration in lung adenocarcinoma and their clinical significant: A gene-expression-based retrospective study [published online ahead of print, 2020 Jan 31]. J Cell Biochem. 2020;10.1002/jcb.29667. doi:10.1002/jcb.29667

Validity of the findings

No comment.

Additional comments

Lung adenocarcinoma (LUAD) is the most common histological type of lung cancers, which is the primary cause of cancer-related mortality worldwide. Tumor microenvironment (TME) cells are important elements in constitutions of tumor tissue. There is increasing evidence that they have important clinical pathological significance in predicting tumor clinical outcomes and therapeutic effects. It is a good idea to investigate the correlation of TME related genes signature with LUAD prognosis, and it might provide new ideas for the diagnosis and treatment of LUAD. In this manuscript, Dr. Chen and colleagues focuesed on the role of the "106 TME related SurGenes" in diagnosis and prognosis of LUAD. The study provided the conclusion that "106 TME related SurGenes" might be used to predict the prognostic survival of LUAD. Although the current study is interesting, there are some major concerns that need to be addressed.

Major concerns:
1. In the newly published article [1], Dr. Chen and colleagues also analysis LUAD dataset from TCGA using ESTIMATE, and the results are similar to this manuscript, so it is necessary to discuss the similarities and differences between the two studies.
[1] Chen G, Dong Z, Wu D, Chen Y. Profiles of immune infiltration in lung adenocarcinoma and their clinical significant: A gene-expression-based retrospective study [published online ahead of print, 2020 Jan 31]. J Cell Biochem. 2020;10.1002/jcb.29667. doi:10.1002/jcb.29667
2. “Data acquisition” Line 99. How to calculate or obtain the immune and stromal scores from ESTIMATE of 517 LUAD samples in TCGA dataset. The detailed method should be provided.
3. “Data acquisition” Line 95. Whether the 442 LUAD samples in GSE68465 were analyzed by ESTIMATE? Please clarify this point.
4. “Construction of the risk assessment model” Line 131 and 139. The “SurFactors” and “Risk socres” are not a common calculation method. Please provide relevant references.
5. "ESTIMATE algorithm of LUAD" Line 156. Please explain the relationship between Estimate score, Immune score and Stromal scores.
6. "Overall survival analysis of DEGs" Line 201. Why did the author to choose the survival curves of CD3G, LAMA2 and ITGAL for analysis subjects? Please explain the meaning in the research.
7. "PPI network construction of prognostic genes" Line 206. What was the basic rationale for “first three important modules”? Please state clearly and provide references.
8. "PPI network construction of prognostic genes" Line 208. What was the basic rationale for "hub genes"? Please state clearly and provide references.
9. "Functional enrichment analysis of significant modules" Line 213-219. What is the meaning of this point? Is there any supporting eveidence for this conclusion?
10. "Prognostic genes validation" Line 228. “...there are 47 genes of particular concern because they have not been proved to be related to the prognosis of LUAD.” What is the basis of this statement? Please state clearly and provide references.
11. “Construct and validation the prognostic model” Line 236. “…the final prognostic model combined with 106 candidate gene were constructed…” 106 TME related SurGenes is might the most important innovation point of this research. Thus, Multivariate Cox regression analysis should be preformed between age, gender, TNM stage and Risk score in this model. The author may refer to this article:
[1] Chen G, Dong Z, Wu D, Chen Y. Profiles of immune infiltration in lung adenocarcinoma and their clinical significant: A gene-expression-based retrospective study [published online ahead of print, 2020 Jan 31]. J Cell Biochem. 2020;10.1002/jcb.29667. doi:10.1002/jcb.29667
12. In figure 1 to 8, organization of all charts needs to be rearranged. The charts should be align and sufficient resolution. The content of main text should be corresponding with the results in charts.

Minor comments:
1. P-values should be marked in a uniform way (Italic capitals), and just keep three decimal places.
2. "ESTIMATE algorithm of LUAD" Line 160. Why did not analyze the relationship between Estimate score and prognosis?
3. "ESTIMATE algorithm of LUAD" Line 174. "transcriptional microarray analysis"?
4. "Identification of DEGs" Line 179. How many DEGs were up regulated in both analysis for comparison based on immune scores nor stromal scores? The data should be shown by Wynne chart.
5. "Overall survival analysis of DEGs" Line 197. "...overall survival <30..." What is the unit, month or year?
6. "PPI network construction of prognostic genes" Line 205. What is "prognostic DEGs"? Please state clearly.
7. Figure 1A to C. A scatter plot should show the quartiles.
8. In figure 1D and E, the range of Y-axis (Percent survival) should be the spectrum from 0 to 100, and the unit of survival time should use “months”.
9. Figure 6,The displaying format of the survival curve of gene PRPRC is different from that of other genes, please keep in consistency.

Reviewer 2 ·

Basic reporting

n/a

Experimental design

n/a

Validity of the findings

n/a

Additional comments

In this manuscript titled "Tumor microenvironment related novel signature predict lung adenocarcinoma survival" by Chen et al. have used gene expression datasets of lung adenocarcinoma patients to identify genes linked with patient survival. They also performed functional annotations to further refine the genesets to generate a prognostic model for survival prediction. Manuscript in the current form is weak and needs additional work to support the hypothesis to further strengthen the manuscript. Following are the issues that need to be addressed.

Major comments:

1. What is the significance of using tumor microenvironment for predicting prognosis?
How does this model differ from other prognostic models for LUAD? A detailed comparison of available prognostic models of LUAD is required with the current model to know the prognostic performance.

2. Authors have shown that immune and stromal scores are predictive of OS. Are they also predictive of progression free survival (PFS)? Fig1 D pls. mention how many samples were used to calculate stats and mean survival stats. TCGA has many cohorts of LUAD (see cbioportal) are these trends similar across various cohorts?

3. How cutoffs for immune and stromal scores were derived? Figure 2 is not informative unless authors highlights the upregulated and downregulated pathways or genes and scale the heatmap. what is the significance of each cluster? Instead of using all DEGS authors should use only highly variable one to make this heatmap meaningful. There is no legend to show the expression range. Expression values should be normalized for better visualization.

4. Its not clear when authors have shown that Estimate stromal and immune scores are predictive of OS, what is the point of using DEG derived from these scores to again perform survival analysis when Estimate scores can also predict? Does a combined DEG score predicts OS better than ESTIMATE Scores?

5. Quality of figures is low and either they are not labelled well or they are not informative. For eg. what is the significance of PPI maps in Figure5? why these nodes are highly connected?

6. how many DEGS were overlapping b/w immune vs stromal scores?

7. introduction and discussion section of the manuscript is poorly written and lacks clear questions and findings of the paper. Figure 3 should be combined with figure 2 and all the pathway enrichments should be mapped on the heatmap
why string database?

8. fig 1 : how p values were derived
9. define AJCC
10. line 175: its RNA-seq not microarray

·

Basic reporting

The authors' hypothesis the TME in LUAD indicated by ESTIMATE is associated with patients' survival. However, this manuscript is quite similar to one previous study (Profiles of immune infiltration in lung adenocarcinoma and their clinical significance: A gene‐expression–based retrospective study. Chen et al, doi.org/10.1002/jcb.29667). Although authors performed a pathway annotation analysis on the key genes of TME, I don't think it adds additional significant value in this field.

Experimental design

This experimental design is acceptable but there are a few missing key information.
1. How did the authors get the ESTIMATE score? Did they get the score through the ESTIMATE website or by performing ESTIMATE on LUAD locally?
2. How were the groups determined in the differential gene analysis? What are the groups?
3. In survival analysis, authors can not utilize the student t-test as statistical analysis for the Kaplan-Meier curve and Cox regression. (line 119)

Validity of the findings

More details should be addressed. Here are some examples of blurred statement.
1. In figure 1 (A-C), which groups did the p-value represent for? Did the author perform ANOVA between all groups? Then, a post-hoc test should be considered.
2. In figure 1D, I don't believe there is a statistical significance between these two groups.
3. In Cox regression, authors should present a table listing the Cox coefficients and significance. Authors also should address why they chose CD3G, LAMA2, and IGTAL in figure 4

Additional comments

Although the authors proposed an interesting topic in this field, the authors seem not to provide novel results or insights.
Some details of the design should be provided.
Some English grammar needed to be improved.

---

## Round 0.2 · Major Revisions

After checking the manuscript I consider it still needs a major revision.

The methodology is not clear in many points (see at the annotated pdf). In the manuscript you talk about "low" and "high" expression groups, but you do not define the values in the text (e.g. line 250).

I strongly suggest to check the text by a native speaker. I have found a lots of sentences which were hard to understand.

I also emphasise to add Chen et al, doi.org/10.1002/jcb.29667 for comparison, due to the use of ESTIMATE score on TCGA datasets.

The reviewers mention statistical errors regarding the figures. Also please do not use screen shots and make sure the figures quality better in general.

Please use consistent language throughout the manuscript e.g. Stromal score vs stromal score. Please also add the acronyms e.g. RMA,
NOM, STRING.

Finally, you found CD19, a B-Lymphocyte Surface Antigen and Receptor-type tyrosine-protein phosphatase C which is involved in T-cell receptor activation. Please check what is their role in LUAD, if any. Please elaborate in the discussion much more thoughtfully what your analysis means regarding LUAD treatment/parthenogenesis.

Reviewer 1 ·

Basic reporting

No comment.

Experimental design

No comment.

Validity of the findings

No comment.

Additional comments

Lung adenocarcinoma (LUAD) is the most common histological type of lung cancers, which is the primary cause of cancer - related mortality worldwide. Tumor microenvironment (TME) cells are important elements in tumor tissue. There is increasing evidence that they have important clinical pathological significance in predicting tumor clinical outcomes and therapeutic effects. It is a good idea to investigate the correlation of TME related genes signature with LUAD prognosis, and it might provide new ideas for the diagnosis and treatment of LUAD. In this manuscript, Dr. Chen and colleagues looked at the role of the "106 TME related SurGenes" in diagnosis and prognosis of LUAD. The study provides the conclusion, which showed that "106 TME related SurGenes" were expected to be used to predict the prognostic survival of LUAD. The manuscript is improved and most concerned raised by the reviewer have been addressed, but there is still room for improvement in the figures. In order to facilitate the readers and increase the influence of the article, please continue to improve this article.

Minor comments:
1. Figure 1A to C. A scatter plot of population data needs to show the median and quartile values. The “bars” should be black instead of the same color as the “dots”.
2. Figure 1D to E. For the convenience of the reader, the survival curve takes months.
3. Figure 3. For clarity, use vector graphics instead of direct screenshots.
4. Figure 4. Please enlarge the network structure diagram. The text in the picture must be clearly visible.
5. Figure 5C and D. For the convenience of the reader, the survival curve takes months. Mark the sample number of each group.
6. Figure 6. Please enlarge the GSEA results. The text in the picture must be clearly visible.
7. Figure 7. Please enlarge the figures. The text in the picture must be clearly visible.
8. Figure 8A and E. For the convenience of the reader, the survival curve takes months. For the convenience of the reader, the survival curve takes months. Mark the sample number of each group. Please unify the format of all survival curves, including units of time, ordinate amplitude, and markup format!
9. Figure 9. For clarity, use vector graphics instead of direct screenshots.

Reviewer 2 ·

Basic reporting

NA

Experimental design

NA

Validity of the findings

NA

Additional comments

Figure 2: Gene expression heatmap should be z- normalized across patient samples for better representation.

Figure 2 C: WHat is the p value of overlap?

Figure 3: What does color of bar indicates?

Figure 4: These PPI maps are not informative unless the nodes names are marked

Figure 5: A-B It is noyt clear how these pvalue are derived. If it is one pvalue - it should have been derived from ANOVA rather than wilcoxon rank sum test.

Figure 6: Instead of so many enrichment plots which is unreadable, only selected GSEA should be shown.

Figure 7: It is not clear why a correlation value of 0.001 i.e NO correlation is also significant?

---

## Round 0.3 · accepted · Accept

First of all, I would like to apologise for the late reply on my side. After checking the manuscript I agree with the reviewer that it is ready for publication.

Reviewer 1 ·

Basic reporting

No comment.

Experimental design

No comment.

Validity of the findings

No comment.

Additional comments

Lung adenocarcinoma (LUAD) is the most common histological type of lung cancers, which is the primary cause of cancer - related mortality worldwide. Tumor microenvironment (TME) cells are important elements in tumor tissue. There is increasing evidence that they have important clinical pathological significance in predicting tumor clinical outcomes and therapeutic effects. It is a good idea to investigate the correlation of TME related genes signature with LUAD prognosis, and it might provide new ideas for the diagnosis and treatment of LUAD. In this manuscript, Dr. Chen and colleagues looked at the role of the "106 TME related SurGenes" in diagnosis and prognosis of LUAD. The study provides the conclusion, which showed that "106 TME related SurGenes" were expected to be used to predict the prognostic survival of LUAD. The paper is improved and most concerned raised by the reviewer have been addressed. I think it is might suitable for publication at this version of revised manuscript.